# A Comparison of Visual Quality and Contrast Sensitivity between Patients with Scleral-Fixated and In-Bag Intraocular Lenses

**DOI:** 10.3390/jcm11102917

**Published:** 2022-05-22

**Authors:** Yueh-Ling Chen, Christy Pu, Ken-Kuo Lin, Jiahn-Shing Lee, Laura Liu, Chiun-Ho Hou

**Affiliations:** 1Department of Ophthalmology, Chang Gung Memorial Hospital, Taoyuan 333, Taiwan; yuehling20@gmail.com (Y.-L.C.); d12093@cgmh.org.tw (K.-K.L.); leejsh@cgmh.org.tw (J.-S.L.); 2Department of Medicine, School of Medicine, Chang Gung University, Taoyuan 333, Taiwan; 3Institute of Public Health, School of Medicine, National Yang Ming Chiao Tung University, Taipei 112, Taiwan; cypu@nycu.edu.tw; 4Department of Ophthalmology, National Taiwan University Hospital, Taipei 100, Taiwan

**Keywords:** visual quality, contrast sensitivity, scleral fixation, intraocular lens

## Abstract

Purpose: To analyze visual quality and contrast sensitivity in patients after intraocular lens (IOL) implantation with sutured scleral fixation. Setting: Chang Gung Memorial Hospital, Taoyuan, Taiwan. Design: Retrospective observational study. Methods: Data on the refractive outcome, visual acuity, and subjective visual symptoms in patients with scleral-fixated or in-bag IOL implantation were collected from September 2019 to March 2020. We also investigated patients’ postoperative higher-order aberrations (HOAs) and dysphotopsia using a wavefront aberrometer and glaretester, respectively. The following values were compared: corrected distance visual acuity, spherical equivalent, root mean square values for aberrations, and contrast sensitivity. Results: A total of 23 eyes implanted with scleral-fixated IOL and 74 eyes with in-bag IOL were studied. The mean postoperative spherical equivalent and logarithm of the minimum angle of resolution after scleral fixation were −1.09 ± 3.32 D and 0.20 ± 0.17, respectively. The ocular HOAs were higher in the scleral-fixation group than in the in-bag group (*p* = 0.001). Contrast sensitivity was negatively associated with age, and it was similar between the two groups after controlling for the age effect. Conclusions: Ocular HOAs and refractive errors were higher in the scleral-fixation group than in the in-bag group. However, no significant difference was noted in contrast sensitivity between advanced scleral fixation and in-bag IOL implantation.

## 1. Introduction

Cataract surgery is the most common ocular surgery, with more than 20 million procedures performed worldwide [1]. Cataract surgery serves as both a visual restoration operation and refractive procedure [2]. Typically, the intraocular lens (IOL) is placed in a capsular bag. However, patients who experience posterior capsule rupture, zonular dialysis, dropped lens, or dislocated IOL during trauma or ocular surgery may receive alternative techniques such as anterior chamber, iris-fixated, or scleral-fixated IOLs [3].

The scleral-fixated IOL, first mentioned by Malbran [4] in 1986, has become a popular technique for patients with inadequate capsular support. The advantage of IOL scleral fixation over anterior chamber IOL implantation is the reduced risks of corneal endothelial loss, peripheral anterior synechiae, cystoid macular edema, and hyphema [5]. Although scleral-fixated IOL implantation has the problem of suture exposure, modified techniques can be applied to cover the suture ends with scleral pockets [6]. Another method is sutureless intrascleral-fixated IOL. A three-piece IOL is inserted into the anterior chamber, and the haptics are pulled out and positioned in the scleral tunnels. However, the complication of this technique includes intraoperative haptic breakdown [3].

Visual acuity can be maintained or improved with scleral fixation [7], but visual quality with regards to higher-order aberrations (HOAs) and contrast sensitivity remain issues. The IOL tilt and decentration after scleral fixation are greater than those after in-bag implantation [8]. IOL decentration can lead to dysphotopsia [9], and IOL tilt induces a considerable amount of ocular coma-like aberrations [10,11]. The appropriate positioning of an IOL is crucial to satisfactory visual quality following cataract surgery. However, the literature on dysphotopsia and contrast sensitivity after scleral-fixated IOL surgery is scant.

The purpose of this study was to compare the visual acuity, aberrometry, and glare disability of eyes treated with scleral fixation with those of eyes treated with standard cataract surgery.

## 2. Methods

### 2.1. Patients

This retrospective study comprised patients who underwent standard cataract surgery or transscleral fixation of the IOL. Inclusion criteria were as follows: patients with pseudophakia with in-bag or scleral-fixated IOLs, 20 years or older, no complications during IOL implantation, no ocular disorders such as severe non-proliferative diabetic retinopathy (NPDR) or PDR, corneal opacities or epithelial defects, severe macular degeneration or dystrophy, optic atrophy, amblyopia in the operated eye, or posterior capsule opacification after cataract surgery that could degrade visual quality. Indications for scleral fixation included aphakia and subluxation or dislocation of the crystalline lens or IOL. Only monofocal IOL implantation was studied because multifocal IOLs would have introduced a confounding effect with respect to dysphotopsia. We recruited patients between September 2019 and March 2020. Patients who could not undergo examination as a result of dementia or mental disorders were not included. All eyes had a minimum postoperative time of 1 month when the inflammation subsided without postoperative steroid use, corneal edema, or anterior chamber reaction, to ensure the wound and visual acuity were stable. We excluded patients with postoperative corrected distance visual acuity (CDVA) of more than 0.5 logarithm of the minimum angle of resolution (logMAR) because they could lose contrast sensitivity [12]. The study adhered to the tenets of the Declaration of Helsinki, and it was approved by the Institutional Review Board of Chang Gung Memorial Hospital, Taoyuan, Taiwan (Approval number: 2101220033). Written informed consent was waived because of the retrospective nature of the study.

### 2.2. Postoperative Ophthalmic Examinations

The CDVA was measured, and slit-lamp biomicroscopy, contrast sensitivity testing, pneumatic tonometry, indirect ophthalmoscopy, and aberrometry were performed. The postoperative CDVA was converted to logMAR values and compared between the scleral-fixation group and the in-bag group.

### 2.3. Contrast Sensitivity Test

Contrast sensitivity was evaluated with best refractive correction without pupil dilation using a CGT-2000 contrast glaretester (Takagi Seiko, Takaoka, Japan). Contrast sensitivity testing was performed under daytime (100 cd/m^2^), twilight (10 cd/m^2^), and nighttime (5 cd/m^2^) luminance conditions with and without glare at a test distance of 5 m. The area under the log contrast sensitivity function (AULCSF) was calculated for statistical analysis [13].

Postoperative perceptive dysphotopsia was assessed using a questionnaire, with a point given for each category. Subjective photic phenomena, including glare, halo, starburst, and coma, were evaluated with a penlight held 1 m in front of the tested eye under mesopic conditions at the outpatient department. Symptoms were rated as 0 = none, 1 = mild, 2 = moderate, or 3 = severe. Additionally, this questionnaire was filled out by the nurse. Higher mean scores indicated less satisfactory results. The mean score for each category was calculated and tested for significance.

### 2.4. Optical Aberrations

Wavefront measurements were postoperatively obtained using a refractive power and corneal analyzer (OPD-Scan III, NIDEK, Tokyo, Japan). This device used the fundamental principle of automatic retinoscopy, and it provided integrated corneal topography and wavefront measurement. The retina was scanned with a slit-shaped light beam, and the reflected light was captured by an array of rotating photodetectors over a 360° area. The aberrometer offered an aberration profile of the whole eye, and the root mean square values for aberrations, HOAs, tilt, coma, spherical aberrations, trefoil, and astigmatism were measured for statistical analysis. Wavefront maps were analyzed with a 3-mm pupil diameter up to the fourth-order Zernike coefficients. The pupil sizes were also measured by this wavefront aberrometer under mesopic condition.

### 2.5. Surgical Technique

Mydriasis was achieved preoperatively with 1% tropicamide eyedrops and 10% phenylephrine eyedrops. Sutured scleral fixation was performed by an experienced surgeon (LL), and phacoemulsification and in-bag IOL implantation were done by another (CHH). Scleral fixation was conducted using the four-point fixation technique described by Khan et al. [14], with some modifications. A 2.65-mm transparent corneal incision was made after retrobulbar anesthesia, and the IOL was loaded in the injector and injected into the anterior chamber. The two haptics were looped with a 10-0 polypropylene suture intraocularly at the nasal sclera 2 mm posterior to the limbus. The same step was repeated on the temporal side. Additional procedures such as vitrectomy and IOL exchange may have been performed at the time of scleral fixation. The standard phacoemulsification surgery was performed using the following procedures under topical anesthesia: clear corneal incision of 2.65 mm, continuous curvilinear capsulorrhexis with an approximate diameter of 5.0 mm, hydrodissection, phacoemulsification, irrigation and aspiration, and in-bag IOL implantation using an injector.

### 2.6. Statistical Analysis

All statistical analyses were performed using Stata, version 15 (StataCorp, College Station, TX, USA). Independent *t* tests were employed to compare the visual quality between the two groups. Generalized estimate equation method (GEE) was performed to identify factors affecting contrast sensitivity, which was set as the dependent variable (AULCSF). The following parameters were included as explanatory variables: age, sex, pupil size, surgical technique (scleral fixation or in-bag IOL implantation), IOL type (spherical, aspheric, or toric), logMAR, and ocular aberrations. Another GEE was conducted to determine the factors affecting subjective dysphotopsia (glare, halo, starburst, and coma), with the same aforementioned explanatory variables. A *p* value of less than 0.05 was considered statistically significant.

## 3. Results

This study comprised 100 eyes from 70 patients. Three eyes from three patients were excluded because the CDVA was greater than 0.5 logMAR. A total of 97 eyes from 67 patients were analyzed, of which 23 eyes underwent sutured scleral fixation, with a mean patient age of 58.13 years (range 36–79 years), and 74 eyes underwent in-bag IOL implantation, with a mean patient age of 69.76 years (range 46–96 years). A total of 15 eyes (20.27%) of 10 patients had diabetes mellitus in the in-bag group, while 6 eyes (26.09%) of 5 patients were affected in the scleral-fixated group. One patient in the in-bag group had Sjogren’s syndrome. The mean spherical equivalent was −1.09 ± 3.32 D in the scleral-fixation group and −0.23 ± 0.75 D in the in-bag group. The CDVA of the in-bag group was slightly better (mean logMAR 0.11 vs. 0.20). Statistically significant differences were observed between the two groups in terms of age, sex, IOL type, CDVA, and spherical equivalent (Table 1). Although 33 eyes (34%) were followed within 3 months after the surgery, the mean CDVA could reach 0.16 logMAR. In the scleral-fixation group, 14 eyes had IOL subluxation or dislocation; 4 eyes had lens dislocation, and 5 had aphakia following complications with cataract extraction. The majority of sutured IOLs in the scleral-fixated group (56.52%) received Akreos Adapt Advanced Optics lenses (Bausch + Lomb, Laval, QC, Canada).

The postoperative wavefront data including ocular, internal, and corneal aberrations for both groups are listed in Table 2. Ocular aberrations differed markedly from corneal aberrations between the scleral-fixation and the in-bag group in terms of overall aberrations, HOAs, tilt, coma, trefoil, and astigmatism, with the exception of spherical aberrations.

In terms of contrast sensitivity testing, the in-bag group performed better under daytime luminance conditions with or without glare interference but similar to those in the scleral-fixation group under twilight or night conditions (Table 3). For the in-bag group and scleral-fixation group, 53 (71.62%) and 11 eyes (47.83%) were tested for perceptive dysphotopsia, respectively. The mean questionnaire scores for subjective dysphotopsia under mesopic conditions with glare showed no significant differences between the two groups. Symptoms of glare, halo, coma, and starburst were similar in both groups (*p* > 0.5).

The results of GEE for contrast sensitivity are described in Figure 1 and Table 4. Age had significantly negative effects on contrast sensitivity under photopic and mesopic conditions, and the surgical technique (in-bag and scleral fixation) did not affect the results of contrast sensitivity after controlling for the age effect. LogMAR and ocular aberrations had significantly negative effects on contrast sensitivity under every luminance condition with or without glare interference. No significant variable was determined in the GEE for dysphotopsia.

## 4. Discussion

This study demonstrated that the mean CDVA, spherical equivalent, and ocular HOAs were significantly better in the in-bag group than in the scleral-fixation group. However, in the contrast sensitivity test, no difference was noted between the groups except under photopic conditions, which was compatible with the result of the subjective dysphotopsia questionnaire. The factors related to low-contrast sensitivity were age, logMAR, and ocular aberrations.

Mimura et al. [15] reported that the mean spherical equivalent was –1.16 ± 2.28 D for transscleral-fixated IOL implantation at 2 years, and Mizuno et al. [16] indicated that the mean postoperative CDVA in the logMAR at 1 month was 0.25 ± 0.41. Both results were similar to those of our study. Huang et al. [17] determined that IOL scleral fixation induced an average 1.66 D myopic shift, which may be caused by the more anterior placement of the scleral-fixated IOLs [16]. Hayashi et al. [8] demonstrated that anterior chamber depth with sutured IOLs was shallower than that with in-bag IOLs, which caused a significant myopic shift. Other studies [18,19,20,21] have also reported an increase in spherical equivalent in those who underwent IOL scleral fixation.

Most of our patients who received transscleral IOL fixation were men, and they were much younger than those who underwent in-bag IOL implantation. The main reason for scleral fixation was trauma experienced during labor work, and the majority of laborers were men. One study [15] with a 12-year follow-up noted that the mean patient age after scleral fixation was 61.7 years, and another study [10] recorded more men in their scleral-fixation group than in the in-bag group (44.4% vs. 41%), which was similar to our study.

Ocular coma aberration was significantly greater in the scleral-fixation group than in the in-bag group. A study [10] indicated that IOL tilt correlated with ocular coma-like aberrations. Therefore, an increase in ocular coma aberrations in the scleral-fixation group in our study suggests the contribution of an IOL tilt. Ocular trefoil aberration was also greater after scleral fixation. Torii et al. [22] noted the same results and reported that postoperative ocular, corneal, and internal trefoil-like aberrations were significantly greater in their scleral-fixation group than in the intracapsular group. Spherical aberration was correlated with the implanted IOL type, and aspheric IOLs were associated with lower spherical aberrations than spherical IOLs. Our results demonstrated that the percentage of aspheric IOLs, including those that were toric, in the scleral-fixation group was comparable to that in the in-bag group (74.32% vs. 60.87%, *p* = 0.21), indicating no difference in ocular spherical aberration between the two groups.

In this study, we observed that patients with scleral-fixated IOLs had worse contrast sensitivity only under photopic conditions, which may be attributable to poor visual acuity and more ocular aberrations in this group. Additionally, visual acuity may predict contrast sensitivity. Rubin et al. [23] reported a linear regression with a correlation coefficient of −0.56 for the logMAR and contrast sensitivity in patients with cataracts. Another study [24] demonstrated consistent results, with a significant correlation between the logMAR and contrast sensitivity (*r* = −0.55). Contrast sensitivity was also affected by ocular aberrations in this study. Many studies [25,26] have reported that deteriorated contrast sensitivity is related to increased HOAs in eyes that underwent keratorefractive surgery. For cataract surgery, however, no studies have indicated such results. Our result suggested that an increase in ocular aberrations contributed to the loss of contrast sensitivity in pseudophakic eyes, which has never before been published. Research into contrast sensitivity in patients with scleral-fixated IOLs was scarce. Gao et al. [27] concluded that IOL decentration and a tilt less than 0.5 mm and 5°, respectively, did not affect postoperative contrast sensitivity under dim light conditions. The results of perceptive dysphotopsia tested in our study also indicated no significant difference between these two groups under mesopic conditions. In the multivariate regression, surgical technique did not significantly affect the extent of subjective glare disability.

Most patients in the scleral-fixation group received Akreos IOLs four-point fixation. Compared with traditional two-point scleral fixation, this technique has a low risk of IOL tilt and decentration. In addition, cystoid macular edema and glaucoma were less commonly observed with four-point fixation [28]. With regard to other complications after suture-fixated IOL procedures, only one suture exposure was noted in our study, which was markedly low compared with a report indicating a 6–27% probability of suture-related complications in transscleral-sutured IOL surgery [5].

Our study had several limitations. First, the sample size of the scleral-fixation group was not large, but the subjective and the objective refraction were both consistent with those of other studies. Second, in this study, we could not evaluate the preoperative visual function because many of our patients were referred from other medical facilities for IOL fixation, and we could not assess the extent of visual improvement following surgery in the scleral fixation group. The causes for inferior CDVA in scleral fixation group could be our fixation method or prior insults, such as trauma or a complication of cataract surgery. This is intrinsic, and it is not possible to differentiate. However, our result showed that with an advanced scleral fixation technique, patients subjectively did not suffer from worse contrast sensitivity than patients with in-bag IOL implantation. Third, several different IOL types were used in our study, providing different spherical aberrations of IOLs, which affected ocular spherical aberrations [2,29,30,31]. However, the percentage of spherical or aspheric IOLs in these two groups was similar, as were the ocular and the internal spherical aberrations. In our multivariate generalized estimating equation analysis, the IOL type (spherical, aspheric, or toric) had no significant effect on contrast sensitivity. Fourth, the patients in the scleral-fixated group were younger than those in the in-bag group, and age was reported to have a negative effect on contrast sensitivity [32,33]. Our result also demonstrated that patients with older age were associated with worse contrast sensitivity. However, after controlling for the age effect, the surgical technique (in-bag and scleral fixation) did not affect the results of contrast sensitivity. Finally, instead of using commercially available dysphotopsia questionnaires [34], we utilized a newly designed method of evaluating dysphotopsia under simulated nighttime luminance. The strength of our study was the simultaneous investigation of objective contrast sensitivity and subjective glare disability following transscleral-sutured IOLs procedures, both of which provided consistent results.

In summary, our study demonstrated that although the logMAR, spherical equivalent, and higher-order aberrations were greater following scleral fixation, there was no significant difference between scleral-fixated IOLs and in-bag IOL implantation in terms of the visual quality and the contrast sensitivity under mesopic conditions. The age, logMAR, and ocular aberrations had negative effects on contrast sensitivity in patients with pseudophakia under different luminance conditions. It is noteworthy that perceptive dysphotopsia under a dim light was similar between these two groups, which means the patients’ satisfaction was comparable, and it is as convincing as the objective measurements when evaluating visual quality. In addition, our study suggested that careful manipulation by a well-experienced surgeon could provide satisfactory outcomes in patients who received transscleral-sutured IOLs with four-point fixation. However, refinement of suturing techniques in scleral fixation is still required to reduce ocular aberrations and to preserve contrast sensitivity under daylight conditions. Further studies should include a larger sample size, sutureless technique, and matched case-control study design, such as age, sex, and visual outcome, to deepen our understanding of the visual quality of IOL scleral fixation.

## Figures and Tables

**Figure 1 jcm-11-02917-f001:**
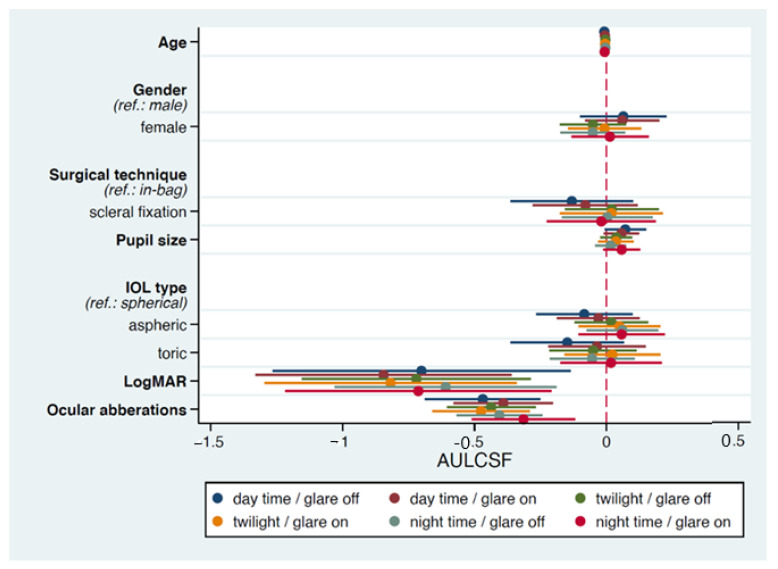
Forest plot depicts the coefficient of each factor and its 95% confidence interval (CI) in a multiple regression analysis for factors associated with contrast sensitivity. The x-axis represents the reference line (dashed), the value of coefficient (dot), and 95% CI (strip). AULCSF = area under the log contrast sensitivity function; IOL = intraocular lens; LogMAR = logarithm of the minimum angle of resolution.

**Table 1 jcm-11-02917-t001:** Demographics and visual outcomes of eyes following in-bag and scleral-fixated intraocular lens implantation.

Characteristics	In-Bag (*n* = 74)	Scleral Fixation (*n* = 23)	*p* Value
Age in years, mean ± SD (range)	69.76 ± 9.58 (46–96)	58.13 ± 12.81 (36–79)	<0.001 *
Male sex, No. (male %)	24 (32.43)	10 (43.48)	0.038 *
Laterality, No.			0.509
OD (%)	36 (48.65)	13 (56.52)	
OS (%)	38 (51.35)	10 (43.48)	
IOL type, No.			0.003 *
Spherical (%)	19 (25.68)	9 (39.13)	
Aspheric (%)	28 (37.84)	14 (60.87)	
Toric (%)	27 (36.48)	0 (0)	
Postop in months, mean ± SD (range)	19.88 ± 30.09 (1–132)	16.22 ± 19.27 (1–60)	0.585
Pupil size (mm ± SD)	4.81 ± 1.02	4.66 ± 0.99	0.526
CDVA (logMAR ± SD)	0.11 ± 0.14	0.20 ± 0.17	0.015 *
SE, mean ± SD	−0.23 ± 0.75	−1.09 ± 3.32	0.0385 *
Astigmatism, mean ± SD	−0.81 ± 0.58	−1.11 ± 0.87	0.0654

CDVA = corrected distance visual acuity; IOL = intraocular lens; logMAR = logarithm of the minimum angle of resolution; OD = right eye; OS = left eye; SD = standard deviation; SE = spherical equivalent; * *p* < 0.05.

**Table 2 jcm-11-02917-t002:** Postoperative ocular, corneal, and internal aberrations for 3-mm pupil diameters in eyes following in-bag and scleral-fixated intraocular lens implantation.

	In-Bag (*n* = 74)	Scleral Fixation (*n* = 23)	*p* Value
Ocular			
Aberrations ^a^ (μm ± SD)	0.387 ± 0.194	0.785 ± 0.667	<0.001 *
HOAs (μm ± SD)	0.145 ± 0.070	0.255 ± 0.241	0.001 *
Tilt (μm ± SD)	0.103 ± 0.064	0.187 ± 0.203	0.003 *
Trefoil (μm ± SD)	0.121 ± 0.076	0.209 ± 0.189	0.002 *
Coma (μm ± SD)	0.035 ± 0.024	0.058 ± 0.053	0.004 *
Astigmatism ^b^ (μm ± SD)	0.434 ± 0.211	0.704 ± 0.553	0.001 *
Spherical aberration (μm ± SD)	0.016 ± 0.013	0.022 ± 0.020	0.059
Corneal			
Aberrations ^a^ (μm ± SD)	0.377 ± 0.183	0.432 ± 0.194	0.213
HOAs (μm ± SD)	0.112 ± 0.049	0.146 ± 0.084	0.019 *
Tilt (μm ± SD)	0.133 ± 0.102	0.167 ± 0.148	0.208
Trefoil (μm ± SD)	0.074 ± 0.042	0.102 ± 0.066	0.017 *
Coma (μm ± SD)	0.056 ± 0.038	0.065 ± 0.060	0.355
Astigmatism ^b^ (μm ± SD)	1.201 ± 0.828	1.340 ± 0.739	0.473
Spherical aberration (μm ± SD)	0.025 ± 0.024	0.035 ± 0.030	0.094
Internal			
Aberrations ^a^ (μm ± SD)	0.404 ± 0.204	0.714 ± 0.683	0.001 *
HOAs (μm ± SD)	0.130 ± 0.066	0.209 ± 0.234	0.011 *
Tilt (μm ± SD)	0.138 ± 0.110	0.178 ± 0.177	0.198
Trefoil (μm ± SD)	0.086 ± 0.061	0.151 ± 0.173	0.007 *
Coma (μm ± SD)	0.051 ± 0.041	0.053 ± 0.037	0.830
Astigmatism ^b^ (μm ± SD)	0.858 ± 0.685	1.340 ± 1.815	0.060
Spherical aberration (μm ± SD)	0.025 ± 0.024	0.035 ± 0.030	0.094

HOAs = higher-order aberrations; * *p* < 0.05; ^a^ lower and higher-order aberrations included; ^b^ astigmatism and secondary astigmatism included.

**Table 3 jcm-11-02917-t003:** Postoperative contrast sensitivity of eyes following in-bag and scleral-fixated intraocular lens implantation.

(AULCSF ± SD)	In-Bag (*n* = 74)	Scleral Fixation (*n* = 23)	*p* Value
Day (100 cd/m^2^)			
Glare off	1.345 ± 0.395	1.073 ± 0.549	0.011 *
Glare on	1.314 ± 0.333	1.065 ± 0.521	0.008 *
Twilight (10 cd/m^2^)			
Glare off	1.179 ± 0.317	1.046 ± 0.434	0.114
Glare on	1.060 ± 0.356	0.887 ± 0.479	0.065
Night (5 cd/m^2^)			
Glare off	1.085 ± 0.316	0.972 ± 0.371	0.157
Glare on	0.783 ± 0.359	0.658 ± 0.443	0.173

AULCSF = area under the log contrast sensitivity function; * *p* < 0.05.

**Table 4 jcm-11-02917-t004:** Multivariate generalized estimating equation (GEE) analysis of contrast sensitivity under every luminance condition with or without glare interference.

Variables		Day Time Glare off	Day Time Glare on	Twilight Glare off	Twilight Glare on	Night Time Glare off	Night Time Glare on
		β (SE)	β (SE)	β (SE)	β (SE)	β (SE)	β (SE)
Age		−0.008 (0.004) *	−0.008 (0.004) *	−0.005 (0.003)	−0.007 (0.004)	−0.007 (0.003) *	−0.009 (0.004) *
Gender		0.080 (0.082)	0.095 (0.076)	−0.027 (0.070)	0.013 (0.073)	−0.027 (0.069)	0.030 (0.077)
Surgical technique		−0.127 (0.113)	−0.145 (0.097)	−0.012 (0.085)	−0.014 (0.096)	−0.039 (0.082)	−0.053 (0.102)
Pupil size		0.069 (0.039)	0.035 (0.034)	0.007 (0.029)	0.014 (0.033)	−0.024 (0.028)	0.032 (0.035)
IOL type	Aspheric	−0.077 (0.091)	−0.007 (0.082)	0.097 (0.074)	0.077 (0.080)	0.122 (0.072)	0.075 (0.084)
	Toric	−0.126 (0.107)	−0.015 (0.093)	0.036 (0.081)	0.059 (0.092)	0.040 (0.078)	0.042 (0.097)
LogMAR		−0.698 (0.277) *	−0.700 (0.235) **	−0.590 (0.200) ***	−0.776 (0.234) **	−0.406 (0.191) **	−0.677 (0.248) *
Ocular aberrations		−0.460 (0.106) ***	−0.387 (0.090) ***	−0.417 (0.078) ***	−0.458 (0.090) ***	−0.405 (0.075) ***	−0.333 (0.095) ***

IOL = intraocular lens; logMAR = logarithm of the minimum angle of resolution; SE = standard error; * *p* < 0.05; ** *p* < 0.01; *** *p* < 0.001.

## Data Availability

Not applicable.

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
