# Peer review of "A Comparison of Visual Quality and Contrast Sensitivity between Patients with Scleral-Fixated and In-Bag Intraocular Lenses"

_jcm, 2022, doi:10.3390/jcm11102917_

Round 1

Reviewer 1 Report

The paper is interesting and  well written.

Could the authors explain:

  • how long after the surgery the refractive outcomes were obtained? Did all the patients included had the same mean follow-up?
  • did the difference between the mean age of the patients of the two group could influnce the visual et refractive outcomes?
  • did the of presence of macular or optic nerf diseases was take in count ?
  • could you mention in the introduction the other techiniques of sutureless iol implantation ?

Reviewer 2 Report

Hou et al. compared visual quality and contrast sensitivity between patients with scleral-fixated IOL and in-bag IOL. A total of 23 eyes implanted with scleral-fixated IOL and 74 eyes with in-bag IOL were studied. They found ocular HOAs and refractive errors were higher in the scleral-fixation group than the in-bag group. No significant difference was noted in contrast sensitivity between advanced scleral fixation and in-bag IOL implantation under mesopic conditions. This paper provides some valuable clinical observation data for scleral-fixated IOL. However, I also have some concerns about participants inclusion:

  1. The age difference between the two groups of scleral-fixated IOL andin-bag IOL is significant, which may lead to the deviation of statistical results. Is it possible to reduce the participants of in-bag IOL group to match the participants of the other group, so that the age difference is not significant?

  1. Were the underlying chronic diseases of immunity, metabolism or others investigated of the participants? Because some chronic diseases may be involved in the course of cataract, they may have an impact on the statistical results.

Reviewer 3 Report

Chen et al. analyze visual outcomes of scleral-fixated IOLs in comparison to in-the-bag placement. The authors do an adequate job presenting their data and recognizing their limitations, and the topic is relatively scarce in the literature. I have the following comments/suggestions regarding the work:

  • Title: I would suggest using a title that reflects the comparative design of the study.
  • As the authors recognize, eyes that receive scleral-fixation IOLs are ones with prior complications/pathology (e.g. trauma, complicated cataract surgery) that might originally limit the visual potential of the eyes. Changes in CSF seen in the results are related - at least in part - to difference in CDVA. Since the authors do not have the preop data on VA, CSF or HOAs, there is no way of knowing if the postop difference is related to the IOL implantation method or to inherent differences between the implanted eyes in each group.
  • The use of both eyes of the same individual may confound the results and is controversial in such design. A generalized estimate equation method would be useful to determine intrapersonal correlation between both eyes. 
  • The inconsistency of the IOL make used for implantation is another important confounding factor.
  • The significant age difference between both groups is a confounding factor that may limit the analysis of variables such as dysphotopsia questionnaire results and contrast sensitivity.
  • The range of follow up in both groups starts at 1 month. This may be insufficient for complete resolution of inflammation and stabilization of IOL position and patient's refraction.
  • Was IOL decentration/tilt assessed in any form? It would be useful to correlate its degree to the HOAs and CSF.
  • Even if both groups had significantly different objective measurements regarding CSF and HOAs, patients in both groups reported similar levels of subjective dysphotopsia symptoms. This should be highlighted in the discussion since patient-perceived outcome is more important than measurements and numbers when assessing visual quality.

Round 2

Reviewer 3 Report

The changes made are satisfactory.